

# Epigallocatechin gallate protects MC3T3-E1 cells from cadmium-induced apoptosis and dysfunction via modulating PI3K/AKT/mTOR and Nrf2/HO-1 pathways

Fanhao Wei[1,2,*], Kai Lin[3,*], Binjia Ruan[3], Chaoyong Wang[3], Lixun Yang[1,2], Hongwei Wang[3] and Yongxiang Wang[1,2]

[1] Clinical Medical College, Yangzhou University, Yangzhou, China
[2] Northern Jiangsu People's Hospital Affiliated to Yangzhou University, Yangzhou, China
[3] Nanjing University Medical School, Nanjing, China
[*] These authors contributed equally to this work.

Corresponding authors
Hongwei Wang, hwang@nju.edu.cn
Yongxiang Wang,
wyx918spine@126.com

## ABSTRACT

Epigallocatechin gallate (EGCG), an active constituent of tea, is recognized for its anti-cancer and anti-inflammatory properties. However, the specific mechanism by which EGCG protects osteoblasts from cadmium-induced damage remains incompletely understood. Here, the action of EGCG was investigated by exposing MC3T3-E1 osteoblasts to EGCG and $CdCl_2$ and examining their growth, apoptosis, and differentiation. It was found that EGCG promoted the viability of cadmium-exposed MC3T3-E1 cells, mitigated apoptosis, and promoted both maturation and mineralization. Additionally, $CdCl_2$ has been reported to inhibit both the phosphoinositide 3-kinase/protein kinase B/mammalian target of rapamycin (PI3K/AKT/mTOR) and nuclear factor erythroid 2-related factor 2/heme oxygenase-1(Nrf2/HO-1) signaling pathways. EGCG treatment attenuated cadmium-induced apoptosis in osteoblasts and restored their function by upregulating both signaling pathways. The findings provide compelling evidence for EGCG's role in attenuating cadmium-induced osteoblast apoptosis and dysfunction through activating the PI3K/AKT/mTOR and Nrf2/HO-1 pathways. This suggests the potential of using EGCG for treating cadmium-induced osteoblast dysfunction.

## INTRODUCTION

Cadmium, a ubiquitous environmental contaminant, poses significant health risks as documented by extensive research (*He et al., 2022*). Cadmium exposure has been linked to various organ impairments, including neurological inflammation, hepatic dysfunction, reproductive toxicity, immune system suppression, and disrupted glucose metabolism (*Borné et al., 2014*; *Chen et al., 2019*; *Huang et al., 2021*; *Liu et al., 2022*). Notably, the toxic effects of cadmium are seen mostly in the kidneys and bones (*Handl et al., 2019*; *Kim et al., 2018*) and short-term exposure can inhibit bone formation (*Ma et al., 2021a*) and

contribute to osteoporosis by inducing cellular senescence (*Luo et al., 2021*). Specifically, cadmium reduces the expression of bone-formation-associated genes in mesenchymal stem cells (MSCs) (*Knani et al., 2019*; *Lv et al., 2019*; *Wu et al., 2020*). Furthermore, cadmium disrupts the bone remodeling process by promoting osteoblasts apoptosis (*Ou et al., 2021*). Despite these findings, the precise molecular mechanism by which cadmium impacts osteoblast formation remains unclear.

PI3K/AKT signaling is implicated in various human diseases, and previous studies have identified it as the classical apoptotic signaling pathway (*Bi et al., 2023*; *Ding et al., 2023*; *Tang et al., 2022*). mTOR, operating downstream of PI3K/AKT, regulates various signaling pathways, including those relevant to therapeutic strategies for osteoarthritis (*Cui et al., 2023*; *Xu et al., 2021b*). More importantly, research has found that cadmium can induce apoptosis of small intestine cells *via* PI3K/AKT/mTOR (*Zhang et al., 2022*). Oxidative stress is a pivotal factor in many pathological processes, including cytotoxicity induced by cadmium (*Geng et al., 2019*; *Zhang et al., 2021*). Research efforts have focused on the functions of transcription factors in the mitigation of damage caused by oxidative stress. In this context, Nrf2 is prominent as it controls the expression of antioxidant genes, such as HO-1. This regulatory pathway thus protects against diseases associated with oxidative stress, including osteoporosis (*Tian et al., 2019*; *Waza et al., 2018*). Recent reports have highlighted the essential roles of the PI3K/AKT/mTOR and Nrf2/HO-1 axes in bone cell metabolism (*Jin et al., 2020*; *Li et al., 2024*; *Wang et al., 2020*; *Zhou et al., 2023*).

Epigallocatechin gallate (EGCG), a flavonoid-3-ol polyphenol, represents the most potent active constituent among tea polyphenols. Previous studies have demonstrated its antioxidant, anti-inflammatory, and antineoplastic activities (*Gan et al., 2018*; *Liu & Yan, 2019*), showing promising clinical efficacy (*Wei et al., 2019*). Despite its effectiveness in treating osteoporosis and fractures (*Lin et al., 2020*; *Liu et al., 2018*; *Xu et al., 2021a*), the precise pharmacological target of EGCG remains unknown. Furthermore, limited information exists regarding the impact of EGCG on osteoblast damage induced by cadmium exposure and the precise pharmacological target of EGCG. Similarly, EGCG prevents ionizing radiation-induced apoptosis in intestinal epithelial cells (*Xie et al., 2020*). Furthermore, several natural compounds, including EGCG, can protect against the damage of various vascular endothelial cells by regulating PI3K/AKT and Nrf2/HO-1 (*Ajzashokouhi et al., 2023*; *Zhang et al., 2019*; *Zhang et al., 2021*). Therefore, this increases the possibility that the EGCG' protective effect against cadmium-induced osteoblast injury may involve the above pathways. The objective of this study was to evaluate, for the first time, the protective effects of EGCG against cadmium-induced osteoblast (MC3T3-E1) damage and to elucidate the underlying signaling pathways and regulatory mechanisms.

## MATERIALS AND METHODS

### Materials

EGCG ($C_{22}H_{18}O_{11}$, MW: 458.37, >99% purity) was acquired from MedChemExpress (Monmouth Junction, NJ, USA). Cadmium chloride ($CdCl_2$, >99% purity) and $\alpha$-MEM (SH30265.01B) were purchased from Macklin (Shanghai, China) and HyClone (Logan, UT,

USA), respectively. Fetal bovine serum (FSD500) and penicillin/streptomycin (15140122) were obtained from ExCell (Dalian, China) and Gibco (Waltham, MA, USA). TUNEL assay kit (G1504) and alkaline phosphatase (ALP) assay kit (C3206) were from Servicebio and Beyotime (Shanghai, China). Antibodies against $\beta$-actin (AC026), cysteine-aspartic protease-3 (caspase-3) (AC030), cleaved caspase-3 (AC033), B-cell lymphoma 2 (Bcl-2) (AG1225), Bcl-2-associated X protein (Bax) (AF1270), PI3K (AF1966), AKT (AF1777), p-AKT (4060), p-mTOR (AF5869), Nrf2 (12721), HO-1 (AF1333), NAD(P)H dehydrogenase 1 (NQO1) (AF7614), Collagen 1 (Col1) (AF1840), and runt-related transcription factor 2 (Runx2) (12556) were from Cell Signaling Technology (Danvers, MA, USA), while osteopontin (OPN) (YT3467) was obtained from Immunoway (Fremont, CA, USA). Analytical-grade chemical compounds were procured from local reputable suppliers.

## Cell culture

MC3T3-E1 cells, kindly provided by the Shanghai Institutes for Life Sciences Cell Resource Center and were maintained in $\alpha$-MEM medium, containing 10% FBS and penicillin/streptomycin antibiotics at 37 °C in a humid incubator with 5% carbon dioxide.

## Cell viability analysis

To figure out the cytotoxic impact of EGCG on MC3T3-E1 cells, a CCK-8 assay (FUDE, China) was employed. It was attempted to seed cells at a density of $5 \times 10^3$ cells per well particularly in 96-well plates, followed by their exposure to varying concentrations of EGCG (range, 10–100 $\mu$M) and/or CdCl$_2$ (range, 1–100 $\mu$M) for different time intervals (6, 12, and 36 h). Following the respective incubation periods, addition of 10 $\mu$L of CCK-8 reagent to each well was undertaken and allowed to incubate for an additional 2 h particularly at 37 °C in the absence of light. Absorbance readings were thereafter attained at 450 nm through a spectrophotometer. This experiment established the non-toxic concentration range of EGCG for subsequent experiments with MC3T3-E1 cells.

## Osteogenic induction

MC3T3-E1 cells ($1 \times 10^6$/well) were inoculated in 12-well plates and grown to 70–80% confluence, after which replacement of the medium with osteogenic-induction medium was undertaken. This medium consisted of $\alpha$-MEM, supplemented with 10% FBS, 10 mM sodium $\beta$-glycerophosphate, 0.1 $\mu$M dexamethasone, and 50 mg/L vitamin C. EGCG (50 $\mu$M) and CdCl$_2$ (20 $\mu$M) were then added to the respective treatment groups. Media were replaced every three days. Following a seven-day induction phase, cellular mineralization and ALP activities were assessed using the 5-Bromo-4-Chloro-3-Indolyl Phosphate/nitroblue Tetrazolium (BCIP/NBT) staining and an ALP assay kit, respectively.

## Alkaline phosphatase staining

Upon completion of seven-day osteogenic induction, it was attempted to remove the culture media, and thrice rinsing of the cells with PBS was thereafter undertaken. Once fixation (4% paraformaldehyde, 30 min) and further washing in PBS were undertaken, cells were exposed to BCIP/NBT solution for 15 min. The washed cells were then evaluated and imaged using an Olympus DP74 light microscope.

## Western blotting (WB)

Osteoblast protein levels were assessed using WB, as described (*Chen et al., 2021*). Specifically, osteoblasts were harvested, rinsed with PBS, and lysed with RIPA buffer with protease and phosphatase inhibitors. Following centrifugation to remove insoluble material in the lysis, the supernatant was collected.

The protein samples were subjected to SDS-PAGE for separation under reducing conditions, and their transfer to PVDF membranes was subsequently undertaken. It was thereafter attempted to block the utilizing a solution of 5% non-fat milk in TBST for 1.5 h particularly at room temperature, followed by extensive rinsing with TBST. Overnight incubation of the membranes was conducted especially at 4 °C with primary antibodies for probing. Following further washes, the blots were treated with secondary antibodies for one hour at room temperature. Images were captured with BeyoECL Moon Plus Western blotting detection system (Beyotime, Jiangsu, China), and analyzed using ImageJ software.

## TUNEL staining

The TUNEL staining was attempted to conduct as previously described (*Yang et al., 2022*). Following thrice rinsing with PBS, fixation of cells in 4% paraformaldehyde particularly for 15 min was undertaken. Subsequently, their treatment with 0.2% Triton X-100 was carried out, followed by TUNEL staining on the basis of the instructions released by the manufacturer. Imaging of the stained cells was realized through a confocal fluorescent microscope (FV3000) attained from Olympus Corp. Quantification was performed by analyzing five pre-defined fields using ImageJ software in a double-blind manner. Average positive cell ratios were then calculated.

## Statistical analysis

Statistical analyses were performed using GraphPad Prism software (version 8.0). Data are presented as mean $\pm$ standard deviation, derived from at least three independent experiments. For comparisons between more than two groups, one-way ANOVA was utilized, followed by Tukey's post-hoc test to adjust for multiple comparisons. For comparisons between two groups, unpaired two-tailed t-tests were employed. A *P*-value of less than 0.05 was considered statistically significant.

# RESULTS

## Protective effects of EGCG against CdCl$_2$-induced cytotoxicity in MC3T3-E1 cells

EGCG (Fig. 1A), a flavonoid-3-ol polyphenol, was applied to MC3T3-E1 cells at concentrations for durations of 6, 12, and 36 h, respectively. While 20 $\mu$M and 50 $\mu$M EGCG markedly elevated viability ($p < 0.05$) at 36 h, 100 $\mu$M EGCG led to time-dependent cytotoxicity (Fig. 1B). Thus, 10, 20, and 50 $\mu$M EGCG were used for further experiments.

CdCl$_2$ treatment for 36 h caused a dose-dependent decrease in MC3T3-E1 cell viability. Concentration above 20 $\mu$M significantly reduced cell viability by more than 50%, hindering further experiments (Fig. 1C). Therefore, 20 $\mu$M CdCl$_2$ was chosen for subsequent studies.

To delve into the impact of EGCG on mitigating CdCl$_2$-induced cell damage, MC3T3-E1 cells underwent a two-hour pretreatment phase with EGCG before exposure to CdCl$_2$ for
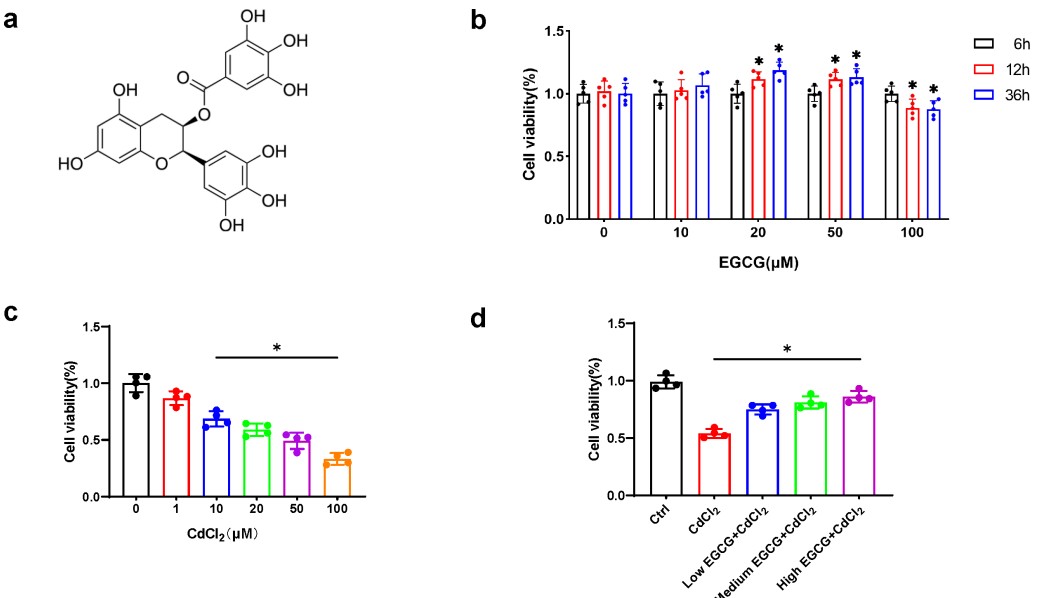

**Figure 1** **EGCG influence on CdCl$_2$-induced viability of MC3T3-E1.** (A) EGCG is a polyphenolic compound. (B) Effects of different concentrations and different treatment times of EGCG on MC3T3-E1 cells viability. (C) Toxic effects within different CdCl$_2$ concentrations exposed for 36 h on MC3T3-E1 cells. (D) EGCG protects MC3T3-E1 cells from CdCl$_2$-induced cytotoxicity. Cells were pretreated with EGCG (10 (low), 20 (medium), 50 (high) $\mu$M) for 2 h and then exposed to 20 $\mu$M CdCl$_2$ for a total of 36 h. Data presentation: mean $\pm$ SD (three independent experiments). * $P < 0.05$ compared with control.

36 h. EGCG treatment ameliorated CdCl$_2$-induced damage in the cells, dose-dependently augmenting cell viability. The most notable enhancement of viability was seen at an EGCG level of 50 $\mu$M (Fig. 1D). Consequently, this concentration was chosen for subsequent experiments.

## Inhibition of CdCl$_2$-induced apoptosis in MC3T3-E1 cells by EGCG

WB was utilized for assessing the levels of key apoptotic markers. Exposure to 20 $\mu$M of CdCl$_2$ significantly increased Caspase-3, Cleaved-Caspase-3 (the activated form), and Bax levels, while decreasing those of Bcl-2 ($P < 0.05$) relative to the controls (Figs. 2A–2E). Additionally, co-treatment with 50 $\mu$M CdCl$_2$ and EGCG significantly reduced the levels of Caspase-3, Cleaved-Caspase-3, and Bax, while that of Bcl-2 was significantly increased compared to the CdCl$_2$-treated group (Figs. 2A–2E). Furthermore, fluorescence microscopy analysis revealed that EGCG treatment attenuated CdCl$_2$-induced apoptosis in the cells. These results suggest that EGCG protects MC3T3-E1 cells from apoptosis induced by CdCl$_2$ (Figs. 2F–2G).

## Protective effects of EGCG against CdCl$_2$-induced dysfunction in MC3T3-E1 cells

To assess whether EGCG could mitigate the effects of CdCl$_2$ on osteoblast formation, we examined the levels of osteoblast markers, such as OPN, Runx2, and Col1, in MC3T3-E1 cells using WB. We found that CdCl$_2$ markedly inhibited osteoblast formation, seen by

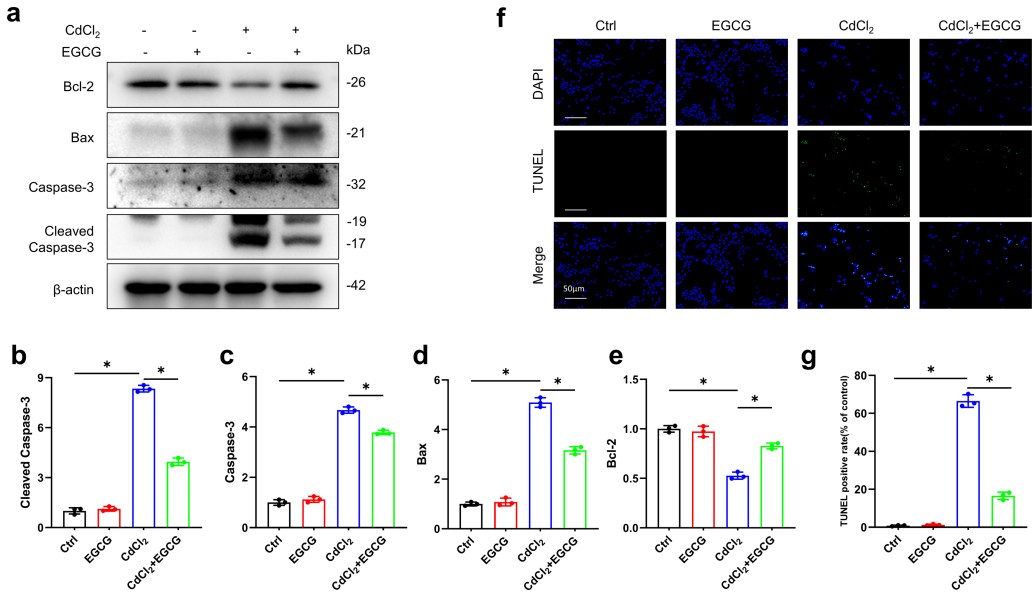

**Figure 2  EGCG influence on CdCl$_2$-induced apoptosis within MC3T3-E1.** (A) Apoptosis-related proteins Cleaved Caspase3, Caspase3, Bax and Bcl-2 protein expressions was determined by immunoblotting. (B-E) Quantitative analysis of Cleaved Caspase3, Caspase3, Bax and Bcl-2 proteins. (F, G) Representative staining of apoptotic cells measured by TUNEL staining (green dots) and DAPI determination of dark background (blue dots) and quantitative analysis of fluorescence. Images were magnified 400 × (horizontal bar = 50 µm). * $P < 0.05$ compared to control.

reductions in Col1, Runx2, and OPN protein levels. However, these effects were markedly reversed after EGCG treatment ($P < 0.05$) (Figs. 3A–3D). ALP (Alkaline phosphatase) staining of osteoblasts confirmed that CdCl$_2$ treatment substantially impeded osteogenic differentiation in the cells ($P < 0.05$). Importantly, EGCG demonstrated a protective effect against reduced osteogenic differentiation due to cadmium treatment (Fig. 3E). These findings collectively suggest that EGCG protects against CdCl$_2$-induced dysfunction in MC3T3-E1 cells.

## Effect of EGCG on PI3K/AKT/mTOR and Nrf2/HO-1 pathways in CdCl$_2$-treated MC3T3-E1 cells

To further investigate EGCG-mediated protection against cadmium-induced damage in osteoblasts and its potential antioxidative properties, we examined key markers in the PI3K/AKT/mTOR and Nrf2/HO-1 pathways using WB. The results demonstrated a marked decrease in PI3K protein following CdCl$_2$ treatment, leading to a significant reduction in AKT and mTOR phosphorylation, while total AKT levels remained unaffected (Figs. 4A–4H). Furthermore, Nrf2 and HO-1 protein levels were also downregulated. Notably, EGCG treatment counteracted these suppressive effects, partially restoring the functions of the PI3K/AKT/mTOR and Nrf2/HO-1 pathways compared to the group treated with CdCl$_2$ (Figs. 4A–4H). These findings suggest that EGCG's protective role likely involves modulating these crucial signaling pathways.

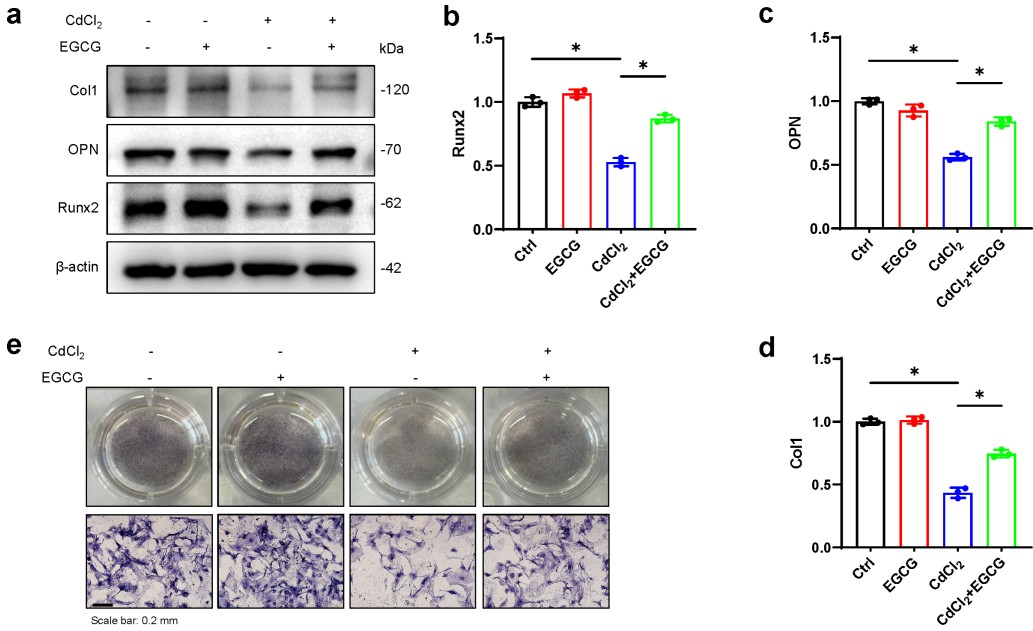

**Figure 3** **EGCG influence on CdCl$_2$-induced dysfunction of MC3T3-E1.** (A) Bone differentiation-related proteins Col-1, NQO1, and Nrf2 protein expressions was detected by Western blot. (B–D) Quantitative analysis of Col-1, NQO1, and Nrf2 proteins. (E) ALP staining of each group after 7 d of osteogenic induction, original magnification × 400. compared with the control group. $^\star$ $P < 0.05$ compared to control.

# DISCUSSION

Here, the mechanisms by which EGCG mitigates CdCl$_2$-induced damage in MC3T3-E1 cells were investigated. The cells were pre-treated with varying concentrations of EGCG (10, 20, and 50 µM) for two hours, followed by CdCl$_2$ exposure for 36 h. The results revealed that high concentrations of EGCG significantly enhanced cell viability, reduced apoptosis by downregulating Caspase-3, Cleaved-Caspase-3 and Bax, and upregulating Bcl-2. Furthermore, EGCG ameliorates the detrimental effects of CdCl$_2$ on osteoblast formation. It has been found that EGCG can promote the expression of osteoprotegerin (OPG) in prostaglandin F2 $\alpha$-stimulated osteoblasts (*Sakai et al., 2017*). Runx2 is a transcription factor associated with bone formation (*Franceschi et al., 2003*; *Hinoi et al., 2006*; *Kim et al., 2023*). Runx2 deficiency in mice, for example, severely disrupts the formation of bone (*Otto et al., 1997*), and its mutations result in clavicular craniosynostosis (*Jin et al., 2023*). EGCG treatment increased the expression of OPN and Col1, established markers of bone differentiation, suggesting its potential to promote osteogenesis (*Dutta et al., 2021*). Our investigation demonstrated that EGCG not only substantially increased Runx2 expression but also antagonized the CdCl$_2$-induced inhibitory effects on osteoblast function and bone formation. These results suggest the involvement of EGCG in the differentiation and metabolic functions of bone.

Cadmium, recognized as a hazardous contaminant, primarily originates from diet, tobacco use, and occupational exposure. It exhibits an extensive elimination half-life in

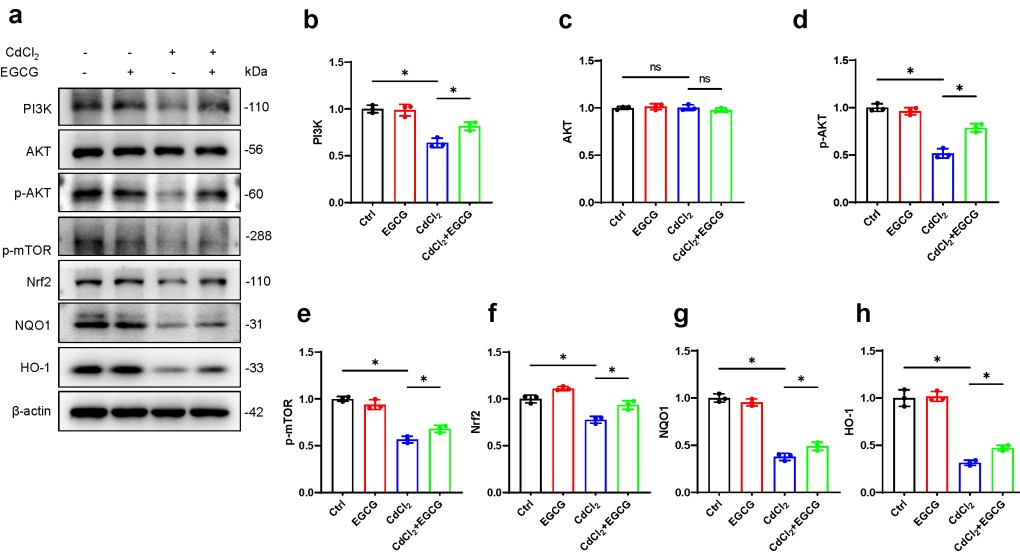

**Figure 4 Investigating EGCG influence on PI3K/AKT/mTOR and Nrf2/HO-1 Pathways in MC3T3-E1 Exposed to CdCl$_2$.** (A) Protein levels of PI3K, AKT, phosphorylated AKT, phosphorylated mTOR, Nrf2, NQO1, and HO-1 were assessed through Western blot analysis. (B–H) Proteins including PI3K, AKT, phosphorylated AKT, phosphorylated mTOR, and Nrf2, NQO1, HO-1 were quantitatively analyzed. Results presentation: mean ± SD, are based on three separate experiments. Significance was noted at *$P <$ 0.05 when compared with the control group.

humans, often decades, and adversely affects the respiratory, renal, reproductive, and skeletal systems (*Yan et al., 2019*). Herbal medications have been found to be effective for the treatment and prevention of bone damage resulting from environmental toxins. These include epimedium, curcumin, and Rhizoma Gastrodiae (*Alam et al., 2022*; *Gao et al., 2012*; *Smirnova et al., 2023*; *Wu et al., 2019*). However, many of these studies lack in-depth investigation of the specific mechanisms involved.

The PI3K/AKT/mTOR pathway regulates diverse physiological and pathological processes (*Fattahi et al., 2020*). Due to its significance, this pathway has become a major focus of research for developing treatment strategies across diverse diseases (*Fakhri et al., 2021*; *Huang et al., 2022*; *Nepstad et al., 2020*; *Yu, Wei & Liu, 2022*). Previous research suggests that PI3K/AKT/mTOR activation promotes osteoblast differentiation in rat BMSC cells (*Zhao et al., 2021*). Moreover, cadmium can induce osteoporosis in ducks *via* the P2X7/PI3K/AKT pathway, which is known to regulate the activity of osteoblasts and osteoclasts (*Ma et al., 2021b*).

In accordance with these findings, we found that EGCG raised PI3K, p-Akt, and p-mTOR levels, along with an elevated Bcl-2/Bax ratio, thus mitigating apoptosis. These results underscore the importance of the PI3K/AKT/mTOR signaling pathway in mediating EGCG's protection against cadmium-induced osteoblast damage.

The Nrf2/HO-1 pathway is a crucial regulator of cellular defenses against oxidative stress, functioning to suppress inflammation and apoptosis (*Meng et al., 2022*). *Xie et al. (2020)* demonstrated that EGCG counteracts the detrimental effects of ionizing radiation on

intestinal epithelial cells by eliminating ROS and preventing cell death. Our study further delved into the antioxidative mechanism of action EGCG. Consistent with recent studies, our findings revealed a notable increase in Nrf2 levels as well as those of the antioxidant enzyme HO-1 in the EGCG-treated group. Our results revealed a possible link between EGCG's anti-apoptotic effect and Nrf2/HO-1 activation. Furthermore, these results suggest that EGCG's protective effect against cadmium-induced damage in MC3T3-E1 cells may involve the activation of both PI3K/AKT/mTOR and Nrf2/HO-1.

## CONCLUSIONS

Our research demonstrates that EGCG protects MC3T3-E1 cells from cadmium-induced cell death and dysfunction. The cytoprotective effects of EGCG may rely on multiple mechanisms, including those associated with the PI3K/AKT/mTOR and Nrf2/HO-1 axes. Our investigation has identified a promising pharmacological agent for treating osteoblast dysfunction induced by cadmium. Further investigation using animal models will explore the mechanism of EGCG in reversing cadmium-induced functional impairment.

**Abbreviations**

| | |
|---|---|
| **EGCG** | Epigallocatechin gallate |
| **CCK-8** | cell counting kit-8 |
| **FBS** | fetal bovine serum |
| **$\alpha$-MEM** | alpha Minimal Essential Medium |
| **MSC** | mesenchymal stem cell |
| **OS** | oxidative stress |
| **OA** | osteoarthritis |
| **ALP** | alkaline phosphatase |
| **P13K** | phosphatidylinositol 3 kinase |
| **Akt** | protein kinase B |
| **mTOR** | mammalian target of rapamycin |
| **Nrf2** | NF-E2-related factor-2 |
| **HO-1** | Heme oxygenase-1 |
| **OPG** | osteoprotegerin |
| **Col1** | collagen 1 |

### Funding

This work was supported by the Jiangsu Province Traditional Chinese Medicine Technology Development Plan Project (ZT202117), the Yangzhou Municipal Key Laboratory Cultivation Project (YZ2021143) and the Second Level Project of the ''333 Project'' of Jiangsu Province (zxkt202203). The funders had no role in study design, data collection and analysis, decision to publish, or preparation of the manuscript.

## Grant Disclosures

The following grant information was disclosed by the authors:

Jiangsu Province Traditional Chinese Medicine Technology Development Plan Project: ZT202117.

Yangzhou Municipal Key Laboratory Cultivation Project: YZ2021143.

Second Level Project of the "333 Project" of Jiangsu Province: zxkt202203.

## Competing Interests

The authors declare there are no competing interests.

## Author Contributions

- Fanhao Wei conceived and designed the experiments, performed the experiments, analyzed the data, authored or reviewed drafts of the article, and approved the final draft.
- Kai Lin performed the experiments, analyzed the data, authored or reviewed drafts of the article, and approved the final draft.
- Binjia Ruan conceived and designed the experiments, analyzed the data, authored or reviewed drafts of the article, and approved the final draft.
- Chaoyong Wang performed the experiments, analyzed the data, prepared figures and/or tables, authored or reviewed drafts of the article, and approved the final draft.
- Lixun Yang performed the experiments, analyzed the data, prepared figures and/or tables, authored or reviewed drafts of the article, and approved the final draft.
- Hongwei Wang conceived and designed the experiments, authored or reviewed drafts of the article, and approved the final draft.
- Yongxiang Wang conceived and designed the experiments, authored or reviewed drafts of the article, and approved the final draft.

## Data Availability

The raw measurements are available in the Supplementary File.

## Supplemental Information

Supplemental information for this article can be found online at http://dx.doi.org/10.7717/peerj.17488#supplemental-information.

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
