# Peer review of "Epigallocatechin gallate protects MC3T3-E1 cells from cadmium-induced apoptosis and dysfunction via modulating PI3K/AKT/mTOR and Nrf2/HO-1 pathways"

_PeerJ, doi:10.7717/peerj.17488_

## Round 0.1 · original submission · Major Revisions

Dear Dr. Wei,

Thank you for your submission to PeerJ!

An expert panel of reviewers have reviewed your manuscripts and offered constructive comments which will improve the quality of the manuscript. I welcome you to submit a revision addressing all the comments raised by the reviewers. In addition, I suggest looking at cleaved caspase 3 in Figure 2A. I look forward to your revision of the manuscript.

Best wishes,
Himangshu

**Language Note:** The review process has identified that the English language must be improved. PeerJ can provide language editing services - please contact us at [email protected] for pricing (be sure to provide your manuscript number and title). Alternatively, you should make your own arrangements to improve the language quality and provide details in your response letter. – PeerJ Staff

Reviewer 1 ·

Basic reporting

no comment

Experimental design

no comment

Validity of the findings

no comment

Additional comments

In the manuscript “Epigallocatechin gallate (EGCG) protects MC3T3-E1 from cadmium-induced apoptosis and dysfunction via modulating PI3K/AKT/mTOR and Nrf2/HO-1 pathways”, the authors found that :1,EGCG impacted cell viability and cytoprotective influence against CdCl2-induced cytotoxicity in MC3T3-E1 Cells; 2, EGCG inhibited CdCl2-induced apoptosis within MC3T3-E1; 3, EGCG protected against CdCl2-induced MC3T3-E1 cell dysfunction; 4, EGCG effected PI3K/AKT/mTOR and Nrf2/HO-1 pathways in CdCl2-affected MC3T3-E1 cells. The authors conclude that EGCG reducs cadmium-induced apoptosis and oxidative stress by activating PI3K/AKT/mTOR and Nrf2/HO-1 pathways.

1,When evaluating EGCG impacts on cell viability, besides CCK-8 assay, flow cytometry is necessary.
2,to evaluate CdCl¢-induced apoptosis, flow is also necessary. Meanwhile, compared with other kind of apoptosis, is there any special markers for CdCl¢-induced apoptosis?
3, the authours shouls expalin why they detected PI3K/AKT/mTOR and Nrf2/HO-1 Pathways but not other pathways? Is there any correlation between CdCl¢-induced apoptosis and I3K/AKT/mTOR and Nrf2/HO-1 ? Why did they examine I3K/AKT/mTOR and Nrf2/HO-1 ? Is there any corelation between I3K/AKT/mTOR and Nrf2/HO-1 under the current setting?
4, the authous investafted the expression of I3K/AKT/mTOR and Nrf2/HO-1. They can concluded the fiding as the effects of EGCG and CdCl¢-induced apoptosis. But they can not concluded it as ‘“via modulating PI3K/AKT/mTOR and Nrf2/HO-1 pathways”’. Please re-consider.

Reviewer 2 ·

Basic reporting

Manuscript title : Epigallocatechin gallate (EGCG) protects MC3T3-E1 from cadmium-induced apoptosis and dysfunction via modulating PI3K/AKT/mTOR and Nrf2/HO-1 pathways
1. The whole manuscript needs excessive English editing.
2. The abstract section needs to be rewritten again. It isn't very clear. The method not found in the abstract section should be added.
3. The introduction should contain background and information relevant to the study.
4. Add the list of abbreviations.
5. The references should be updated.
6. The authors were negligent in writing the whole manuscript for example
Line 36 (Among these, he kidneys and the) and there is many typing errors.
7. line 36-37…. Among these, he kidneys and the skeleton are the main toxic organs for cadmium………….need references
8. the introduction section needs to be rewritten again ,many parapgraphs confused
9. what is the novelty of this research?
10. what is the base for choosing the EGCG at concentrations varying between 10 uM and 100 uM, as well as what is the dose of CdCl2
11. line 110 , what is the specific concentrations of EGCG and CdCl2?
12. line 112, After a 7-day and a 21-day induction phase , what is the state of the cells confluency?
13. Western blotting…… should writing in details
14. for the statistical analysis, why the authors choose the standard deviation not standard errors
15. Figure 1D why the authors compare the result for control only?

16. Biochemical markers analysis should be written in detail with each catalog number of kits
17. It looks like the discussion is in adequate not interpret the whole results

Experimental design

The methods described with insufficient detail & information to replicate.

Validity of the findings

The limitations of the study showed added

Reviewer 3 ·

Basic reporting

The language in this manuscript requires improvement.

Experimental design

The study lacks rigor.

Validity of the findings

The conclusions drawn from the data appear to be overly broad, lacking sufficient compelling evidence.

Additional comments

There are various mistakes in the manuscript as discussed below:

Line 36. It should be ‘THE kidneys’.
Line 39 Please introduce RANKL and its relevance to the story.
Line 52 What is OS?
LINE 146- Correct the sentence- Cells were treated with EGCG.
Line 155- It should be “CdCl2-induced damage”
Figure 1c- X-axis says Cd, It should be CdCl2.
Figure 1d- The labeling needs to be represented in a better way. The cells were pre-treated with EGCG, but the axis only mentions the doses of EGCG. The authors need to add CdCL2 to the panel as well. The cell viability after CdCl2 exposure with EGCG pre-treatment was not restored to the control level even at 50µM. How do the authors explain this?
Figure 2. What is the status of cleaved caspase 3? Activated caspase 3 is a better marker for apoptosis and hence its status needs to be shown. The change in Bax does not look very promising. Further confirmation is needed to show the effects on apoptosis like Annexin-PI staining for flow cytometry etc. Why 50 µM of Cdcl2 is used?
Line 177. The authors need to clarify what ALP is.

I regret to inform you that I cannot recommend the publication of this manuscript. The study's results lack the conclusiveness required to substantiate the effectiveness of EGCG in guarding against cadmium toxicity. The observed findings appear coincidental, lacking substantial evidence of interrelation, such as the PI3K AKT mTOR pathway regulating apoptosis. Comprehensive studies with compelling data are essential to establish the efficacy of EGCG in bone formation.

---

## Round 0.2 · Minor Revisions

Please address the minor concerns of the reviewer #2 and amend your manuscript accordingly.

Reviewer 2 ·

Basic reporting

The authors respond to most of my comments but i still have some concern

1. How the authors determine the sample size of the samples
2. The statistical analysis should be in the details, I am asking about the differences between groups were assessed using ANOVA or t-tests

Experimental design

good

Validity of the findings

good

Additional comments

no

---

## Round 0.3 · accepted · Accept

In my view, all the remaining issues were addressed and revised manuscript is acceptable now